# Evaluating the Enablers of Green Entrepreneurship in Circular Economy: Organizational Enablers in Focus

Maryam Soleimani [1,*], Elahe Mollaei [1], Mojgan Hamidi Beinabaj [1] and Aidin Salamzadeh [2,*]

1   Department of Management, Economics and Accounting, Payame Noor University, Tehran 1599959515, Iran; mollaei@pnu.ac.ir (E.M.); hamidi.mojgan@pnu.ac.ir (M.H.B.)
2   College of Management, University of Tehran, Tehran 141556311, Iran
*   Correspondence: m.soleimani@pnu.ac.ir (M.S.); salamzadeh@ut.ac.ir (A.S.); Tel.: +98-913-265-4162 (M.S.)

**Abstract:** In recent decades, green entrepreneurship has been at the center of attention as an effective strategy to maintain sustainability and create a competitive advantage for organizations in a circular economy. However, the successful implementation of this strategy requires organizations to have internal enablers. This study endeavored to identify and evaluate organizational enablers for green entrepreneurship in manufacturing Small and Medium Enterprises (SMEs) in Iran. Identifying organizational enablers can help SMEs in facilitating the conditions for adopting green entrepreneurship. To these ends, organizational enablers were extracted by reviewing the literature and then, using the viewpoints of 17 active experts in different industries in SMEs, they were classified. In the next step, the "Best Worst Method" was employed to prioritize the identified enablers (5 factors) and sub-enablers (20 factors). The contextual hierarchical relationships between these factors were identified through the "Interpretive Structural Modeling" method. Using the Matrix of Cross-Impact Multiplications Applied to Classification (MICMAC) analysis, the driving and dependence powers of organizational enablers were computed and the enablers were clustered. Based on the results, among the five enablers, three including total quality management, circular supply chain management, and corporate social responsibility were the most important from the point of view of the experts. Moreover, among the sub-enablers, strategic planning, green purchasing, and corporate social responsibility motivation were more important than other sub-enablers. The results of ISM analysis provided a seven-level hierarchical model and the relationships between them. The results of the MICMAC analysis led to the clustering of 20 organizational enablers in three main clusters: driving (nine factors), linkage (four factors), and dependent (seven factors). The results of this study provide practical suggestions for active senior managers to implement green entrepreneurship in SMEs.

**Keywords:** organizational enablers; ISM analysis; BWM; MICMAC

## 1. Introduction

Today, the environment has become an important debate subject that affects different aspects of human life [1]. On the other hand, companies are considered important consumers and polluters of environmental resources. The need to consider environmental issues such as water, air, and solid pollution, deforestation, garbage disposal, decreasing of natural resources, and the need to produce environmentally friendly products has led companies to focus more on green activities in their business [1]. Social awareness of the company's responsibility toward the environment and the growing importance of environmental sustainability in the strategic development of businesses has promoted research related to green entrepreneurship or GE [2]. The literature on GE is still in its early stages of development [3]. As green entrepreneurial companies are recognized as pro-environmental or environmentally friendly or responsible, the way businesses operate in meeting society's needs, while maintaining their ethical responsibility to preserve nature,

is highly sensitive [4]. Having ethical responsibility means that green entrepreneurial companies should have green behavior in their activities. Green behavior refers to paying attention to four green marketing mix factors including green product production, green promotion, green placement, and green pricing [5].

Circular economy (CE) is a regenerative or restorative economic model and an important tool for sustainable development. CE requires changes in sustainable business processes [6]. It is a new concept for organizations that helps them to succeed in sustainable challenges and efficiency of resource consumption [7]. Today, the CE approach has become the focal point of researchers, and experts have considered the urgent need to adopt CE and thus preserve resources as a matter of great importance [8]. Green entrepreneurship is one of the approaches related to CE and is considered a key driver of CE [9]. Green entrepreneurship refers to environmentally friendly businesses that aim to establish solutions for environmental problems while helping communities and promoting economic growth. The concept of GE is expanding in response to the need for sustainable development. Green entrepreneurship provides stability and competitive advantage for companies, especially SMEs [10].

It seems that SMEs play an important role in the management of limited global environmental resources [11]. SMEs are unique assets for development and act as drivers of economic growth and wealth distribution [10]. Accordingly, aligning this important economic sector with positive and effective changes in environmental support activities such as GE and CE can be vital for the economic growth of countries. The challenges that industries face in accepting GE and CE are complicated, as implementing these innovations requires a strong, focused, and unique set of capabilities [12]. What is important is equipping SMEs with GE enablers in the circular economy. Therefore, identifying these enablers for SMEs is crucial before any other action.

Though many studies have been carried out in the green entrepreneurship domain, there are few studies related to how to use organizational enablers to implement green entrepreneurship in the circular economy [12]. Also, the literature review shows that previous studies have not paid attention to the organizational enablers of green entrepreneurship for SMEs in Iran. To fill this research gap, the current study endeavored to identify, prioritize, understand the mutual relationships, and analyze the position of organizational enablers for SMEs in Iran.

By identifying the organizational enablers of CE, this study helps SMEs to evaluate the state of organizational readiness to implement CE. By strengthening the organizational enablers of CE, SMEs can facilitate the conditions of applying CE in a GE. On the other hand, identifying the relationships between these enablers and analyzing the distribution of their driving and dependence powers can help SMEs in strategic planning to strengthen organizational enablers with higher priority.

To meet the goal, the following questions were addressed in a CE and among SMEs:

1. What are the main organizational enablers of GE, and what are their components?
2. How are enablers and sub-enablers prioritized in terms of importance for GE?
3. Which hierarchical structure do enablers follow in their relationships?
4. What position do organizational enablers have relative to each other based on their driving and dependency powers?

## 2. Literature Review

### 2.1. Green Entrepreneurship

Green entrepreneurship is the process of identifying and using entrepreneurial opportunities in such a way as to minimize the negative effects of the company on the natural environment [13]. In other words, GE (or environmental entrepreneurship, eco-entrepreneurship, sustainable entrepreneurship) refers to the implementation of innovations related to environmental protection [3]. Green entrepreneurship is a new branch of entrepreneurship that emphasizes environmental stewardship. Green entrepreneurship combines a strong entrepreneurial spirit with an appreciation for sustainability and other

environmental movements [14]. Green entrepreneurship is rapidly growing with the major goal of integrating the environmental and social benefits of businesses to foster competitive advantages for them [1]. Green entrepreneurship has the potential to become a driving force for a sustainable economic system considering the three fundamental dimensions of society, environment, and economy [15].

### 2.2. Circular Economy

Circular economy (CE) is considered an economic model that aims to effectively use resources to minimize waste, preserve long-term value, improve natural resources, and create closed loops of products and materials in environmental preservation and restoration to achieve socio-economic benefits such as green growth [16]. This concept has emerged as a result of the search for solutions to create more sustainable economies [17]. The goal of CE is to add value to materials and products and achieve maximum lifespan and renewal until the end of their life cycle [18]. CE based on the 3Rs, namely reduce, reuse, and recycle, represents a new commercial alternative to the common linear economy approach based on production–consumption–disposal, and this new approach enables the reduction of resource consumption and waste generation [19]. To be more specific, the CE approach promotes sustainable economic development without creating environmental and resource challenges. CE requires changes in political and economic systems and even changes within individual companies [6]. Implementing CE, economic systems can and should operate based on principles of material, water, and energy cycles in support of natural systems [20]. CE is closely linked to sustainable environmental practices of companies such as improving energy efficiency, using renewable energy sources, and recycling waste, as well as using recycled or renewable materials in raw material supply [6].

### 2.3. Organizational Enablers of GE

Organizational enablers of GE are discussed in the literature through five main factors of circular supply chain management, corporate social responsibility, collaboration, knowledge management, and total quality management. Each of these main factors has several components, resulting in a total of 20 sub-factors (Table 1). The literature related to organizational enablers of GE is presented in the following.

**Table 1.** A summary of the literature on GE enablers.

| | | |
|---|---|---|
| Circular Supply Chain Management | Internal environmental management | [21–24] |
| | Investment Recovery | |
| | Eco-design | |
| | Green information systems | |
| | Green purchasing | |
| Corporate Social Responsibility | Corporate Social Responsibility Motivation | [25–27] |
| | Economic Dimension | |
| | Social Dimension | |
| | Environmental Dimension | |
| Collaboration | Top Management Support | [28–31] |
| | Green Teams | |
| | Cross-Functional Collaboration | |

**Table 1.** *Cont.*

| | | |
|---|---|---|
| | Knowledge Sharing | |
| Knowledge Management Practices | Knowledge Creation | [32–36] |
| | Knowledge Acquisition | |
| | Strategic Planning | |
| | Customer Focus | |
| Total Quality Management | Process Management | [37–44] |
| | Human Resource Management | |
| | Information and Analysis | |

### 2.3.1. Circular Supply Chain Management

Supply chain management (SCM) fosters significant opportunities for empowering business approaches to meet sustainability requirements in the circular economy. Managers in organizations have tried to improve SCM by using optimization techniques [45]. The circular feature of supply chains relates to increasing rates of reuse, remanufacturing, and recycling processes in the relevant economy [46]. In the SCM literature on sustainability, some concepts such as sustainable supply chain, green supply chain, eco-friendly supply chain, and closed-loop supply chain have been introduced and used interchangeably to integrate sustainability concepts into SCM [47]. Circular supply chain management (CSCM) refers to integrating circular thinking into supply chain management and natural and industrial ecosystems around it. In CSCM, technical materials are systematically recovered, and biological materials are regenerated through extensive innovation in business models and supply chain operations, from product/service design to end-of-life management, towards a zero-waste vision. In this process, stakeholders are engaged in the life cycle of a product/service, including component/product manufacturers, service providers, consumers, and users [47]. Research has emphasized the importance of CSCM as an organizational enabler for empowering the CE. For instance, Nguyen et al. confirmed the significant impact of CSCM on environmental sustainability [48]. Momenitabar et al. proposed a mathematical model to design a sustainable bioethanol supply chain network [49,50]. Moreover, Ghasemi et al. proposed a decentralized approach to address production and distribution problems in the supply chain in the hardboard industry [51]. Goodarzian et al. presented a fuzzy sustainable model for a COVID-19 medical waste supply chain network [52]. In another study, Goodarzian et al. presented a model for sustainable supply chains for agricultural products [53]. Safaei et al. presented a new closed-loop supply chain network model to minimize the total network costs [54]. Habib et al. demonstrated that green supply chain activities have a significant positive relationship with the sustainability performance of active textile manufacturing companies in Bangladesh [21]. Green et al. [22] also demonstrated that green supply chain activities, including internal environmental management, green information systems, green purchasing, cooperation with customers, eco-design, and investment recovery, have a significant effect on the organizational performance of manufacturing companies in the United States [23]. In the same vein, Zhu et al. highlighted the positive impact of implementing green supply chain activities on the operational and environmental performance of active automotive companies in China [24]. Moreover, Eltayeb et al. showed that green supply chain initiatives (green purchasing, eco-design, and reverse logistics) have a positive impact on cost reduction and environmental, economic, and intangible consequences [55]. Zhu et al. have also shown that manufacturing companies that implement environmental-oriented supply chain cooperation at a higher level have better performance in terms of CE. In line with the emphasis of the previous research (the importance of CSCM in GE), in this study, this factor is considered as one of the organizational enablers.

### 2.3.2. Corporate Social Responsibility

Corporate social responsibility (CSR) can be defined as the voluntary commitment of companies to contribute to sustainable economic development by integrating social and environmental concerns in their operations and interactions [56]. CSR is a beneficial organizational strategy and an outstanding approach to ethical business and innovation [57]. In recent years, CSR has become an economic necessity in the market, and companies have made efforts to integrate environmental, social, and corporate governance goals in their business models, as well as have increased the development of a more socially responsible decision-making process [19]. A review of previous research showed that CSR has been introduced as one of the enablers for GE or sustainable entrepreneurship. For example, Zeng et al. have introduced social responsibility as a key component in sustainable entrepreneurship [58]. Zeng et al. have shown that social responsibility based on entrepreneurship is effective on the environmental performance of companies with a high level of pollution. Hsu and Chen have also confirmed the relationship between CSR and the environmental performance of the companies [25]. Kraus et al. have shown that CSR can indirectly and through environmental strategy and green innovation have a positive effect on the environmental performance of manufacturing companies in Malaysia [26]. Moreover, Bacinello et al. have demonstrated that corporate social responsibility has a positive effect on sustainable innovation also the business performance of Brazilian companies [27]. Orazalin has also confirmed that companies with effective CSR strategies present better social and environmental performance [27,59]. The results showed that the proactive sustainability strategy (environmental strategy, economic strategy, and social strategy) is effective in the corporate sustainability performance of companies operating in Sri Lanka. Wang et al. [56] have also confirmed that internal and external CSR are effective on disruptive innovation in companies in China. Considering the importance of CSR in GE, this factor is considered as one of the organizational enablers for GE.

### 2.3.3. Collaboration

Improving collaboration (COL) of different actors in product development, process design, and new business models is among the organizational enablers for GE [60]. The importance of COL is that following the CE requires searching for new ways of doing activities. Finding new ways, in turn, requires different perspectives, types of resources and knowledge, and COL between different departments of the organization [28]. Brown et al. have shown that COL between entrepreneurially minded actors strengthens GE in organizations [29]. Lozano has emphasized the importance of COL between actors of societies to achieve sustainable development [30]. This way, it can be concluded that the achievement of organizations to sustainable development and GE also requires COL between actors involved in organizational processes. Raymond Byrne and Polonsky also emphasized that synergy between different departments should be within the organization also among the stakeholders involved in product development and environmentally sustainable delivery processes [31]. Jabbour et al. also confirmed the importance of green teams as key players for companies that intend to improve environmental management practices [32]. Hallstedt et al. have concluded that the superior performance of green product development requires the integration of a sustainable environmental perspective in different parts of the organization and the internal availability of incentives [33].

### 2.3.4. Knowledge Management Practices (KMPs)

Knowledge management (KM) can be defined as a process of strategic organizing to acquire and implement knowledge as an organizational asset that leads to increased performance [34]. The speed and diverse nature of businesses create competitiveness by adapting their knowledge assets that have remained stable over the long term. Knowledge can be defined as dependable information that provides a potential value for the organization [35]. According to the resource-based view approach, organizations with high-level knowledge management processes and enablers have a higher possibility and

ability to produce green and sustainable goods [36]. Previous research has highlighted the importance of knowledge management in innovation and GE. For example, Iqbal and Malik showed that KMPs have a significant effect on the participation of SMEs in sustainable development in Pakistan. Jiang et al. have also shown that knowledge creation (knowledge exchange and knowledge combination) has a positive effect on the company's environmental performance among manufacturing companies in China [37]. Gazali and Zainurrafiqi also concluded that knowledge transfer and integration strengthen the effect of GE on business performance [38]. Based on Chao Wang et al., the knowledge creation process has a positive effect on green products and process innovation [34]. Moreover, they showed that knowledge creation plays a mediating role in the relationship between green entrepreneurial orientation and green innovation. Qader et al. have also shown that KMPs have a significant effect on sustainable entrepreneurship and organizational performance in SMEs in China [39]. Regarding Wong, knowledge sharing has a significant effect on green process innovation from the point of view of green innovation project leaders of manufacturing companies in China [40].

2.3.5. Total Quality Management

Total quality management (TQM) is a management system that focuses on continuous improvement through tools, techniques, and values. The TQM endeavors to increase customer satisfaction by improving the quality of products and services with the least consumption of resources [41]. Quality management significantly improves performance, productivity, and cost reduction and guides sustainable development [42]. TQM and environmental management are related to each other because the goal of TQM is the efficient use of resources, especially natural ones, which is also the main goal of the company's green performance. Moreover, TQM orientation is long-term, considering the impact of organizational activities on the environment and organizational performance in a longer period [41]. In GE, this orientation is at the core of the organization's activities. TQM helps entrepreneurial organizations gain a competitive advantage and create differentiation from competitors and improves their position in the market [43]. Based on the RBV approach, TQM refers to vital, intangible, and unique organizational capabilities, as well as resources that, if implemented effectively, can increase organizational performance and competitive advantage [44]. Previous research has emphasized the importance of TQM to achieve sustainability and GE. For example, Abbas concluded that TQM has a positive effect on the green performance of manufacturing companies in Pakistan. He has confirmed that TQM significantly enhances organizational capabilities to achieve green performance goals. Husnaini, by studying the possible impact of quality management on green innovation and company value in Indonesian companies, also concluded that quality management has a significant effect on green process innovation [42]. Green et al. [22] also studied the effect of TQM on environmental sustainability from the view of managers of manufacturing companies in the US and found that TQM, by reducing waste, also provided environmentally friendly products to customers and had a positive impact on environmental sustainability [61]. Soewarno et al. have also shown that the green innovation strategy, which is one of TQM's activities, is effective in green organizational innovation [62]. Based on Zaid and Sleimi, TQM has a positive effect on improving the business sustainability of production SMEs [44]. Khalil and Muneenam also analyzed the possible impact of TQM on the company's green performance in companies active in the health sector in Pakistan and found that TQM activities have a significant effect on achieving the company's green performance [63].

**3. Methodology**

According to the four questions of the study, a four-step model was developed. In the first step, documentary and Delphi methods were used; in the second step, the best-worst method was used as one of the multi-criteria decision-making methods; in the third step, ISM was conducted as one of the methods of defining one-way/two-way as

well as direct/indirect relationships at different levels; in the final step, MICMAC was implemented as one of the methods of defining the position of variables in independent, dependent, and linkage roles. The steps are as follows:

*3.1. First Stage*

In the first stage, to determine the main organizational enablers of GE in a CE and among small and medium-sized companies, based on searching the keywords sustainable entrepreneurship, environmental entrepreneurship, eco-entrepreneurship, and GE in the databases such as Google Scholar, Science Direct, and Emerald, from the date of 2010 to 2023 and the English language limit, all the articles were collected and their abstracts were reviewed so that the found articles could be passed through the appropriate/inappropriate filter according to the research objectives. After a narrative review of the selected articles, a total of five major factors consisting of 20 sub-components of these articles were selected as organizational enablers of GE in a circular economy.

To finalize the enablers, the two-step Delphi technique was used with the participation of 17 experts (Table 2). The Delphi method was completed in two rounds like this: in the first round, after explaining the research and goals to the experts, the enablers were provided with a semi-structured questionnaire to answer these three questions: (1) Is there any enabler that is not included in this list? (2) Is there any redundant enabler which should be removed? (3) Is the classification of sub-enablers appropriate?

**Table 2.** The characteristics of experts.

| Expert | Job Position | Year of Experience | Gender | Firm Size | Industry Type |
|---|---|---|---|---|---|
| 1 | General manager | 12 | Male | Small | Textile |
| 2 | Production Manager | 6 | Male | Medium | Steel Manufacturing |
| 3 | Marketing Manager | 7 | Female | Medium | Chemicals and Chemical Products |
| 4 | Operations Manager | 9 | Female | Medium | Wood and furniture products |
| 5 | Senior Manager | 16 | Male | Small | Rubber and Plastics Products |
| 6 | Sales Manager | 5 | Male | Small | Household Appliances |
| 7 | Marketing Manager | 8 | Female | Medium | Textile |
| 8 | General Manager | 9 | Male | Small | Rubber and Plastics Products |
| 9 | General Manager | 10 | Male | Small | Medical Equipment |
| 10 | Production Manager | 7 | Male | Small | Chemicals and Chemical Products |
| 11 | Marketing Manager | 12 | Male | Medium | Ceramic products |
| 12 | Operations Manager | 11 | Male | Small | Electric Component Manufacturing |
| 13 | Sales Manager | 7 | Female | Medium | Textile industry |
| 14 | Production and Operation Manager | 9 | Male | Medium | Rubber and Plastics Products |
| 15 | Senior Manager | 17 | Male | Small | Household Appliances |
| 16 | General Manager | 13 | Female | Medium | Petrochemical |
| 17 | Production Manager | 10 | Male | Medium | Wood and furniture products |

In the second round, a checklist containing the main enablers and their components was provided to the experts, and they were asked to give their opinion about their suitability as GE enablers and their classification structure on a 5-point scale (completely suitable, suitable, moderately suitable, slightly suitable, not suitable). The average decision-making criterion of 3.5 was considered so that if the average of each enabler was greater than 3.5, it would remain in the study as a suitable enabler in relation to the research objectives. All the main and secondary enablers obtained this criterion. After collecting the data in the second

round and reviewing the proposed opinions, finally, the enablers and the classification presented in Table 3 were finalized as the result of the first stage.

**Table 3.** Group, local, and global weights and ranks for main/sub-enablers.

| Enablers | "Best" Enabler by Expert | "Worst" Enabler by Expert |
|---|---|---|
| **Main Enablers** | | |
| **Circular Supply Chain Management** | 3, 7, 8, 9, 10, 12, 15, 16 | |
| **Corporate Social Responsibility** | 14 | 10 |
| **Collaboration** | | 2, 4, 5, 6, 9, 11, 13, 17 |
| **Knowledge Management Practices** | | 1, 3, 7, 8, 12, 14, 15, 16 |
| **Total Quality Management** | 1, 2, 4, 5, 6, 11, 13, 17 | |
| **Circular Supply Chain Management** | | |
| **Internal Environmental Management** | 5, 12, 14 | 6, 7 |
| **Investment Recovery** | 6, 8, 10, 11, 13 | |
| **Eco-design** | | 1, 2, 3, 8, 12, 15, 16, 17 |
| **Green Information Systems** | 7 | 4, 5, 9, 10, 14 |
| **Green Purchasing** | 1, 2, 3, 4, 9, 15, 16, 17 | 11, 13 |
| **Corporate Social Responsibility** | | |
| **Corporate Social Responsibility Motivation** | 3, 4, 10, 15, 16 | 7 |
| **Economic Dimension** | | 1, 2, 3, 4, 5, 6, 8, 9, 10, 11, 12, 13, 14, 15, 16, 17 |
| **Social Dimension** | 6, 7, 8, 11, 12 | |
| **Environmental Dimension** | 1, 2, 5, 9, 13, 14, 17 | |
| **Collaboration** | | |
| **Top Management Support** | 1, 2, 3, 4, 5, 6, 7, 8, 9, 10, 11, 12, 13, 14, 16, 17 | |
| **Green Teams** | | 1, 4, 7, 8, 11, 12, 15, 17 |
| **Cross-Functional Collaboration** | 15 | 2, 3, 5, 6, 9, 10, 13, 14, 16 |
| **Knowledge Management Practices** | | |
| **Knowledge Sharing** | 1, 8, 9, 10 | 3, 4, 5, 11, 14, 15 |
| **Knowledge Creation** | 2, 3, 4, 5, 6, 7, 12, 13, 14, 15, 16, 17 | |
| **Knowledge Acquisition** | 11 | 1, 2, 6, 7, 8, 9, 10, 12, 13, 16, 17 |
| **Total Quality Management** | | |
| **Strategic Planning** | 1, 2, 3, 5, 7, 8, 9, 10, 11, 12, 13, 14, 15, 16, 17 | |
| **Customer Focus** | 4 | 11, 15, 16 |
| **Process Management** | | 4, 5, 12 |
| **Human Resource Management** | | 1, 2, 6, 8, 9, 10, 13, 14 |
| **Information and Analysis** | 6, 1 | 3, 7, 17 |

### 3.2. Second Stage

As to the second research question about prioritizing the main and secondary enablers in terms of their importance, the best-worst method [64,65] was used based on independent observations. In this method, each participant is asked to select the best and the worst

(the most effective and the least effective enabler in relation to GE in a CE for SMEs), to determine the values of the two matrices, best-others (BOM) and others-worst (OWM), on a scale of 1 to 9. Weights were calculated through the solver compiled by Rezaei (BWM-Solver-5.xlsx). If the rate of inconsistency in the responses of the experts was more than the threshold, by consulting and revising the weightings, the consistency in the responses would be increased to the minimum acceptable level (Table 4).

**Table 4.** Group, local, and global weights and ranks for main/sub-enablers.

| Main Enablers | | | | Sub-Enablers | | | | Global Weight/Rank | |
|---|---|---|---|---|---|---|---|---|---|
| **Enabler** | **Code** | **Group Weight** | **Group Rank** | **Enabler** | **Code** | **Local Weight** | **Local Rank** | **Global Weight** | **Global Rank** |
| Circular Supply Chain Management | CSCM | 0.285 | 2 | Internal environmental management | IEM | 0.198 | 3 | 0.057 | 7 |
| | | | | Investment Recovery | IRE | 0.239 | 2 | 0.068 | 3 |
| | | | | Eco-design | EDE | 0.085 | 5 | 0.024 | 16 |
| | | | | Green information systems | GIS | 0.151 | 4 | 0.043 | 14 |
| | | | | Green purchasing | GPU | 0.327 | 1 | 0.093 | 2 |
| Corporate Social Responsibility | CSR | 0.186 | 3 | Corporate Social Responsibility Motivation | SRM | 0.315 | 1 | 0.059 | 6 |
| | | | | Economic Dimension | ECD | 0.103 | 4 | 0.019 | 17 |
| | | | | Social Dimension | SDI | 0.291 | 2.5 | 0.054 | 9.5 |
| | | | | Environmental Dimension | END | 0.291 | 2.5 | 0.054 | 9.5 |
| Collaboration | COL | 0.088 | 5 | Top Management Support | TMS | 0.69 | 1 | 0.06 | 5 |
| | | | | Green Teams | GTE | 0.165 | 2 | 0.015 | 18 |
| | | | | Cross-Functional Collaboration | CFC | 0.145 | 3 | 0.013 | 20 |
| Knowledge Management Practices | KMP | 0.094 | 4 | Knowledge Sharing | KSH | 0.283 | 2 | 0.027 | 15 |
| | | | | Knowledge Creation | KCR | 0.563 | 1 | 0.053 | 11.5 |
| | | | | Knowledge Acquisition | KAC | 0.154 | 3 | 0.014 | 19 |
| Total Quality Management | TQM | 0.347 | 1 | Strategic Planning | SPL | 0.367 | 1 | 0.127 | 1 |
| | | | | Customer Focus | CFO | 0.191 | 2 | 0.066 | 4 |
| | | | | Process Management | PMA | 0.131 | 5 | 0.046 | 13 |
| | | | | Human Resource Management | HRM | 0.158 | 3 | 0.055 | 8 |
| | | | | Information and Analysis | IAN | 0.153 | 4 | 0.053 | 11.5 |

### 3.3. Third Stage

To answer the third question (what hierarchical structure does the mutual relations of these enablers follow?), the ISM method was used. This method is efficient especially when there is a complex system in terms of the number of intervening factors and multiple and multilateral relationships. In this method, basic data include the structured self-interaction matrix that is completed by each of the participants using the VAXO rule. At this stage, the

participant needs to evaluate the role of both factors based on 136 pairs of relationships and from the four possible modes (row-to-column effect or V, column-to-row effect or A, row-to-column mutual effect or X, and independence rows and columns from each other or O), to choose a mode to achieve the goal of GE in the context of a CE for SMEs. According to the transitivity principle in the relationships between variables (the simplest of which states that if A is effective on B and B affects C, then A will affect C), there is a possibility that some inconsistencies occur in the responses. To this end, and with an emphasis on reducing inconsistencies, necessary explanations were provided to the experts and the required data were collected.

Finally, the 13 structural self-interaction matrices obtained from the participants were transformed into a structural self-interaction matrix using the mode method (Table 4), which reflects the agreement of the participants about the type of relationship between all pairs of the sub-enablers. The initial accessibility matrix was obtained based on the transformation of VAXO codes into zero and one binary codes (Table 5). The final accessibility matrix (Table 6) was obtained after resolving the inconsistencies in the initial matrix. Based on the final accessibility matrix, stratification factors and their two-way and one-way relationships were formulated in the interpretive structural model. The required calculations were performed using the SIM package in the R workspace.

**Table 5.** SSIM: structural self-interaction matrix.

| | IAN | HRM | PMA | CFO | SPL | KAC | KCR | KSH | CFC | GTE | TMS | END | SDI | ECD | SRM | GPU | GIS | EDE | IRE | IEM |
|---|---|---|---|---|---|---|---|---|---|---|---|---|---|---|---|---|---|---|---|---|
| IEM | A | A | A | A | A | A | A | A | A | A | A | V | V | V | A | X | A | X | X | |
| IRE | A | A | A | A | A | A | A | A | A | A | A | V | V | V | A | X | A | X | | |
| EDE | A | A | A | A | A | A | A | A | A | A | A | V | V | V | A | X | A | | | |
| GIS | X | V | V | X | A | V | V | V | V | V | A | V | V | V | A | V | | | | |
| GPU | A | A | A | A | A | A | A | A | A | A | A | V | V | V | A | | | | | |
| SRM | V | V | V | V | A | V | V | V | V | V | A | V | V | V | | | | | | |
| ECD | A | A | A | A | A | A | A | A | A | A | A | X | X | | | | | | | |
| SDI | A | A | A | A | A | A | A | A | A | A | A | X | | | | | | | | |
| END | A | A | A | A | A | A | A | A | A | A | A | | | | | | | | | |
| TMS | V | V | V | V | X | V | V | V | V | V | | | | | | | | | | |
| GTE | A | X | X | A | A | A | A | A | X | | | | | | | | | | | |
| CFC | A | X | X | A | A | A | A | A | | | | | | | | | | | | |
| KSH | A | V | V | A | A | X | X | | | | | | | | | | | | | |
| KCR | A | V | V | A | A | X | | | | | | | | | | | | | | |
| KAC | A | V | V | A | A | | | | | | | | | | | | | | | |
| SPL | V | V | V | V | | | | | | | | | | | | | | | | |
| CFO | X | V | V | | | | | | | | | | | | | | | | | |
| PMA | A | X | | | | | | | | | | | | | | | | | | |
| HRM | A | | | | | | | | | | | | | | | | | | | |
| IAN | | | | | | | | | | | | | | | | | | | | |

**Table 6.** Level partitioning of GE sub-enablers.

| Sub-Enablers | Driving Power | Dependence Power | Level of Partitioning | Sub-Enablers | Driving Power | Dependence Power | Level of Partitioning |
|---|---|---|---|---|---|---|---|
| IEM | 7 | 20 | 1 | GTE | 11 | 13 | 3 |
| IRE | 7 | 18 | 2 | CFC | 11 | 13 | 3 |
| EDE | 7 | 18 | 2 | KSH | 14 | 9 | 4 |
| GIS | 17 | 6 | 5 | KCR | 14 | 9 | 4 |
| GPU | 7 | 18 | 2 | KAC | 14 | 9 | 4 |
| SRM | 18 | 3 | 6 | SPL | 20 | 2 | 7 |
| ECD | 7 | 20 | 1 | CFO | 17 | 6 | 5 |
| SDI | 4 | 20 | 1 | PMA | 11 | 13 | 3 |
| END | 4 | 20 | 1 | HRM | 11 | 13 | 3 |
| TMS | 20 | 2 | 7 | IAN | 17 | 6 | 5 |

*3.4. Fourth Stage*

To tackle the fourth question (with regard to driving and dependence power, what is the position of organizational enablers in relation to each other?), the MICMAC analysis technique was conducted. In this technique, by analyzing the simultaneous distribution of driving and dependence power and comparing it with the middle of both scales, the position of the variables is determined in terms of independent, dependent, or linking. Accordingly, the position and role of the agents in independent, dependent, linking, or autonomous roles is defined. In the relevant matrix, the sum of the rows was considered as the driving power and the sum of the columns was considered as the final dependence power, and the four-zone diagram of MICMAC was depicted based on them.

## 4. Results

*4.1. Results of BWM Analysis*

With a general look at Table 2 and the distribution of experts' identification codes in different cells, it is possible to reach a kind of consensus regarding the best and worst organizational enablers of GE in a circular economy. Regarding which is the most important organizational enabler among the five main enablers (circular supply chain management, corporate social responsibility, collaboration, knowledge management practices, and total quality management), a bipolar situation can be observed. Eight experts have chosen circular supply chain management and an equal number chose total quality management as the most important organizational enabler. The same bipolar pattern is also observed in relation to the least important enabler (the worst category) in such a way that eight experts have selected collaboration as the least important organizational enabler and the same number of experts have chosen knowledge management practices.

Regarding the most important enabler among organizational sub-enablers, the most agreement is observed in relation to collaboration, total quality management, and knowledge management practices. The vast majority of experts (16 people) have chosen the support of top management as the most important enabler of the collaboration categories. In the same way, a significant majority of experts (13 individuals) have chosen strategic planning as the most important category of total quality management. Finally, the majority of experts (12 individuals) have chosen knowledge creation as the most important organizational enabler for GE in the CE among the categories of knowledge management practices. There is less consensus regarding the enablers of corporate social responsibility and circular supply chain management.

The level of agreement regarding the least important (worst) and secondary organizational enablers is also different. As to the main enablers, as a bipolar was formed for the most important enabler, a bipolar has also been formed in this regard, with a different grouping. While eight experts considered the enabler of collaboration as the least important enabler, the same number of experts considered knowledge management practices as the least important compared to the other four categories.

The highest agreement (16 individuals out of 17) among sub-enablers is related to the economic dimension of the social responsibility category as the least important enabler. After that, the most agreement is related to the enabler of knowledge management practices. In this regard, eleven experts considered knowledge acquisition as the least important category. Regarding other enablers including circular supply chain management, collaboration, and total quality management, less consensus is observed. Regarding the last three enablers, investment recovery (eight individuals), cross-functional collaboration (nine individuals), and human resource management (eight individuals) have been selected by experts as the least important enablers among other enablers of each main category.

The information reflected in Table 4 is based on the averages obtained for the weights of each of the main and secondary enablers. The results revealed that in total, according to the 17 studied experts, among the main enablers, total quality management with a weight of 0.346 and circular supply chain management with a weight of 0.285 are the most important organizational enablers for GE in a circular economy. Meanwhile, collaboration

with a weight of 0.088 and knowledge management practices with a weight of 0.094 are less important compared to other enablers.

Among the sub-enablers for circular supply chain management, green purchasing with a weight of 0.327, for corporate social responsibility, corporate social responsibility motivation with a weight of 0.315, for collaboration, top management support with a weight of 0.690, for knowledge management practices, knowledge creation with a weight of 0.563, and for total quality management, strategic planning with a weight of 0.367 are the most important organizational enablers for GE in a circular economy.

### 4.2. Results of ISM and MICMAC Analyses

From the total of 190 defined paired relationships, couple relationships based on mutual influence include 11.5% of all relationships (X symbols). Moreover, there are no pairs with any one-way or two-way effects (symbol O); this way, 88.5% of the relationships defined by experts are one-way relationships (symbols V or A).

Figure 1 shows the communication mechanism between 20 enablers in a hierarchical manner at seven levels. The Figure can be seen as a roadmap for policymakers and planners who want to move small or medium-sized companies towards green entrepreneurship in a circular economy. The lowest level (first level) includes three factors (environmental, social, and economic dimensions of corporate social responsibility) which are directly or indirectly affected by seventeen other factors, and the highest level (seventh level) includes two factors (strategic planning and senior management support), which directly or indirectly affect all other eighteen factors. In view of this, to determine a starting point for moving towards green entrepreneurship in small and medium-sized companies, according to experts, the seventh level is the best starting point. Considering the domino effect that exists in a hierarchical structural model, it can be expected that if we influence enablers at higher levels such as strategic planning, top management support, corporate social responsibility motivation, customer focus, green information system, information and analysis, the enablers at lower levels such as green purchasing, internal environmental management, investment recovery, eco-design, and environmental, social, and economic dimensions of corporate social responsibility can be affected.

The most driving power is related to the support of top management and strategic planning. Both enablers have a driving power of 20 (the maximum possible). After that, the motivation of corporate social responsibility with a driving force of 18 and green information systems, customer focus, and information and analysis are placed with a driving force of 17. Conversely, internal environmental management and economic, social, and environmental corporate social responsibility have the highest possible dependence power (here 20).

In total and according to the simultaneous distribution of organizational enablers of GE in a CE (Figure 2), nine enablers (including top management support, strategic planning, corporate social responsibility motivation, information and analysis, customer focus, green information systems, knowledge sharing, knowledge acquisition, and knowledge creation) with driving power higher than the middle of the scale and lower dependence power than the middle of the scale were independent variables. Seven enablers (including investment recovery, eco-design, green purchasing, internal environmental management, economic dimension, environmental dimension, and social dimension) with dependence power higher than the average of the scale and driving power lower than the average limit of the scale were dependent variables. Finally, four enablers (including human resource management, process management, green teams, and cross-functional collaboration) with driving and dependence powers higher than the average limit of the scale were identified as linkage enablers.

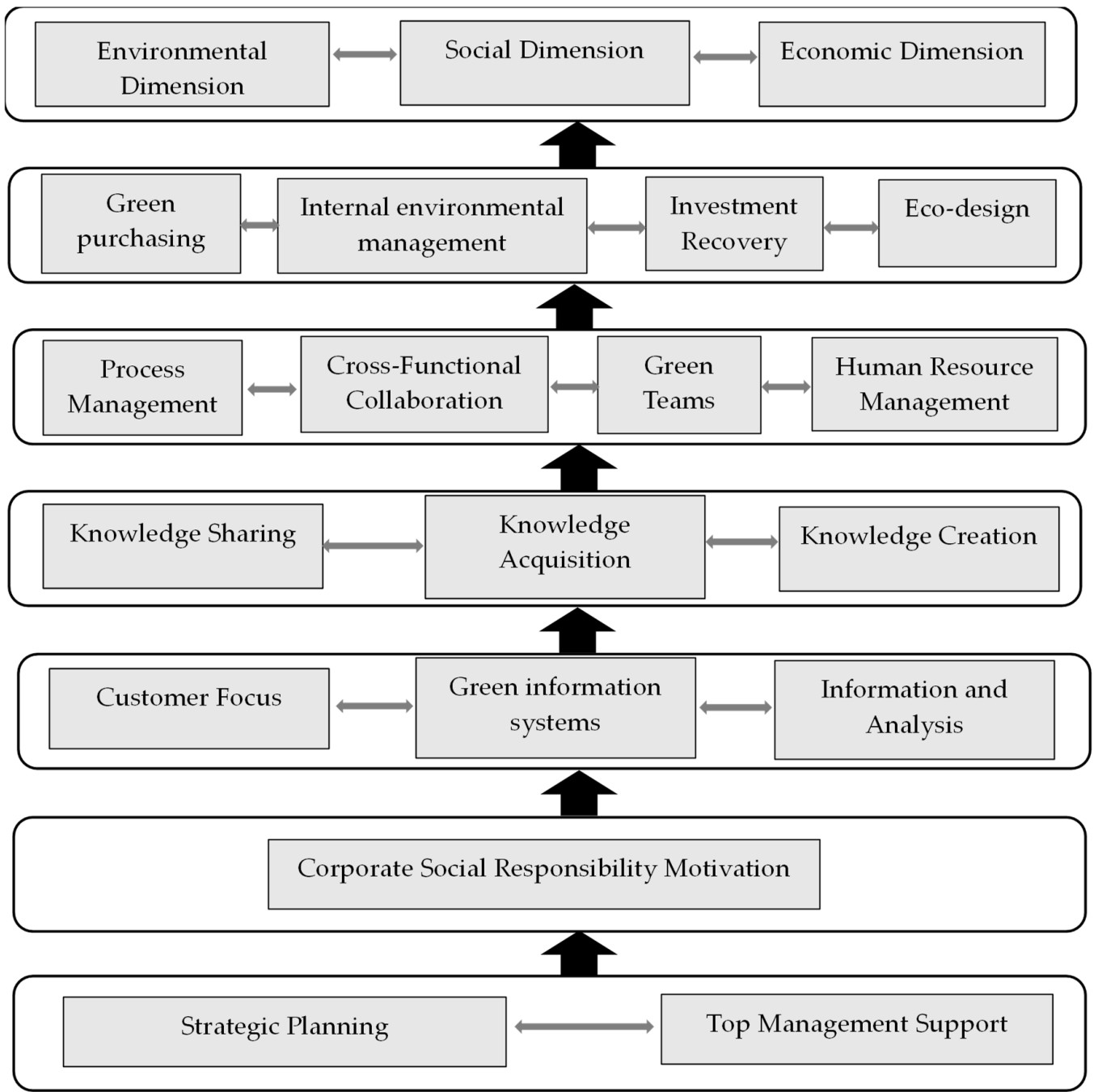

**Figure 1.** ISM-based model of green entrepreneurship.

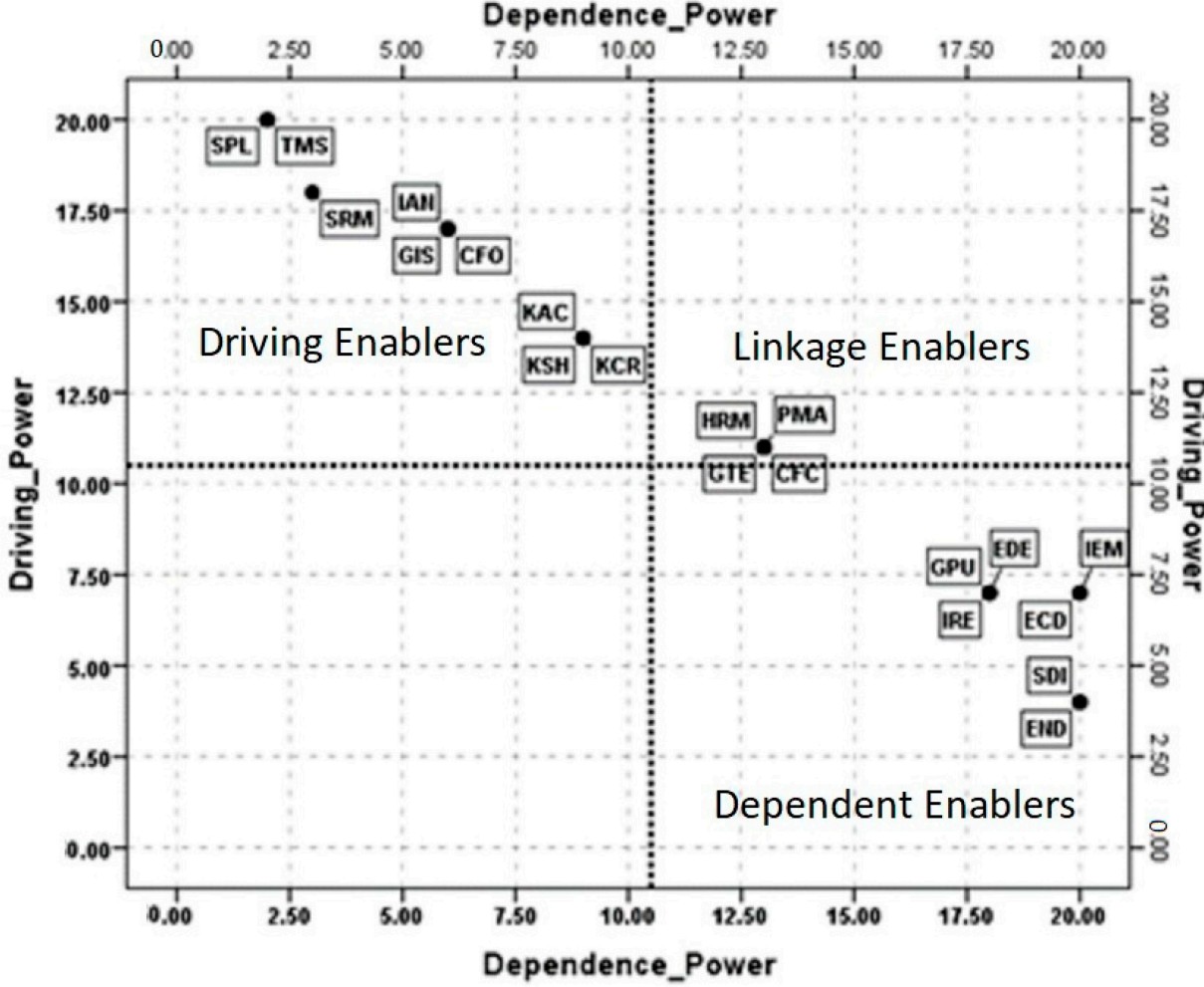

**Figure 2.** Common distribution of sub-enablers based on MICMAC rules.

### 4.3. Validation

Different techniques were used to validate the results at different stages of the research. In the first stage, after determining the main enablers and their components, the two-stage Delphi method was employed. The results of this technique implicitly meant the final confirmation of a two-level structure (main and sub) for enablers. In the second stage, the BWM was used to prioritize and give weight to the enablers. In this method, the inconsistency rate was calculated in the comparisons. In cases the inconsistency rate was higher than the acceptable level, by reviewing pairwise comparisons with experts and modifying them, the rate was reduced to an acceptable level. In the third stage, to validate the results, the model proposed by Sushil [66] was as follows: (1) Regarding the question of whether all relevant elements have been included in the model, a comprehensive review of past studies in the research period provides this assurance. Moreover, because the experts did not propose a new enabler in the Delphi method, it can be concluded that the most important organizational enablers for GE in SMEs in Iran were identified and included in the research. (2) Regarding the question of whether the observed relations were interpreted accurately, the validation of relations was confirmed through the checking of the final model by experts. (3) To analyze the validation and stability of the seven-level structural interpretive model, sensitivity analysis was performed. Sushil suggests sensitivity analysis based on changing one-by-one relationships in the reachability matrix and redrawing the model after every modification. A significant change in the relations of the structural model in each case indicates that the model is sensitive to that change. Under such circumstances, it is necessary to reinterpret the results in those relations with caution. All 136 relations

defined in the reachability matrix were changed separately. In all cases, the number of levels and the relations of the model did not change. Therefore, it can be inferred that the developed structural model is not sensitive to any single relationship. (4) Finally, in response to the question of what the applications of the obtained structural model in authentic settings are, these points can be underlined. Managers of SMEs in Iran could analyze the state of organizational readiness to use GE by evaluating each enabler of the model. The successful adoption of GE in SMEs depends on the favorable condition of each of the organizational enablers. On the other hand, the relationships in the model determine the importance of enablers in terms of priority, which can be considered as a basis for organizational strategic planning. According to the activities carried out to validate the results, it is expected that the level of uncertainty in the obtained model will be reduced to a great extent.

## 5. Discussion

This study endeavored to identify and evaluate the enablers of GE in the CE for manufacturing SMEs in Iran. To this end, the most important enablers and sub-factors related to them were identified using the literature of subject and experts' views. The results of this stage led to the identification of five main enablers and twenty sub-enablers. In the next step, using the BWM method, these enablers were prioritized in terms of their importance. Then, the interrelationships of sub-enablers were identified using the ISM method and finally, a seven-level structural hierarchical model was extracted. In the final stage, by using MICMAC analysis, the position of sub-enablers was analyzed in terms of independent, dependent, or linkage roles. In the following, the details related to the interpretation of the results are provided.

### 5.1. Discussion about BWM Results

The results of the analysis of the most and least important organizational enablers of GE in a CE for SMEs showed that total quality management (TQM) was the most important organizational enabler; that is, the experts participating in the research have agreed on TQM as the most important organizational enabler of GE in SMEs compared to four other main factors, i.e., circular supply chain management (CSCM), corporate social responsibility (CSR), collaboration (COL), and knowledge management practices (KMPs). TQM is a management system compatible with the environment. This approach helps to reduce energy consumption and minimize defects and waste, which is the core of green production measures, by establishing the right production foundations [67]. The desire to minimize waste through TQM practices provides an opportunity to reduce environmental risks for companies [68]. Total quality management has emerged as an important tool to help organizations to achieve sustainable development [35]. Research [41,42,44,61,63] has highlighted the importance of TQM for activities related to GE.

Based on the results, CSCM has taken second place of importance for moving toward GE in manufacturing SMEs. CSCM has focused on achieving sustainable development through the integration of environmentally friendly methods in the traditional supply chain. The traditional supply chain relies on a constant input of virgin natural resources and an unlimited environmental capacity to absorb waste, while CSCM focuses on eliminating or minimizing negative environmental effects (air, water, and land pollution) and resources waste (energy, materials, products) from the stage of extracting raw materials to the use and disposal of final products [20,24,55]. The research [21,23,24,55] emphasized the importance of CSCM as an organizational enabler to achieve GE.

Other results revealed that CSR is ranked third among the main organizational enablers of GE. Commitment to SCR requires that senior managers of organizations purposefully integrate environmental, social, and economic concerns into their business activities and strategies [26]. With a commitment to social responsibility, companies can maintain sustainable operations and growth to achieve sustainable development goals [58]. Orazalin has confirmed that companies with effective CSR strategies show better environmental and

social performance [59]. Strategically implementing CSR will bring benefits to companies and society. In this way, companies achieve an improved brand image and reputation and more profit, while the environmental and social problems of the society will also be reduced [69]. Previous research [25–27,58,59] has emphasized the importance of SCR for implementing GE in organizations.

Knowledge management practices is ranked fourth in terms of importance. Knowledge management is considered a method to increase organizational effectiveness using knowledge and skills [35]. Knowledge management practices include the dimensions of knowledge sharing, knowledge creation, and knowledge acquisition. Using effective KMPs, organizations and entrepreneurs can achieve competitive advantages by improving organizational performance and sustainable entrepreneurship [39]. Past research [35,38–40] has also emphasized the key role of KMPs in providing the conditions for implementing GE in organizations.

Finally, the results of the research revealed that among the main enablers of GE, COL is in the fifth and last position of importance. COL includes the dimensions of top management support, green teams, and cross-functional collaboration. In effect, GE requires COL at all levels of the organization, and achieving complex GE strategies is not possible without collaboration between different actors [28]. Previous research [29–31,33] has also emphasized the importance of this factor for implementing GE in organizations.

Regarding the main factor of TQM, which was identified as the most important main enabler, strategic planning was evaluated as more important than the other four sub-factors (customer focus, process management, human resource management and information and analysis). This way, to implement GE in SMEs, it seems necessary to include its requirements in the mission statement, goals, and strategies of the organization. This leads to the general direction of the organization's activities towards GE. When environmental issues become the main goal of organizational identity, it forces members of the organization to participate more in activities related to the environment [62].

Examining the priority of the sub-enablers of CSCM introduces green purchasing as the most important subcategory. Since green purchasing is the first step in the value chain of an organization, it is considered an essential factor in CSCM [70]. Moreover, the second most important factor in the sub-enablers related to CSCM is investment recovery. This factor requires the sale of excess inventory, waste and used materials, and surplus capital equipment of the organization [23]. Therefore, it can be concluded that recovering the highest value from obsolete products and surplus items can be the foundation of GE in SMEs. The results related to the importance of the sub-enablers of CSR showed that CSR motivation is the most important sub-factor of this organizational enabler. In fact, having the motivation to apply the principles of CSR can encourage managers and employees to perform more work in the field of GE. Examining the priority of KMPs demonstrated that knowledge creation is more important than the other two factors (knowledge sharing and knowledge acquisition). An organization that prioritizes the creation of knowledge and thus innovation, encourages interpersonal exchange of ideas, experiences and challenges established patterns can provide value for customers and society by shaping a vision for GE and the process of creating knowledge [48]. Examining the priority of COL showed that top management support is more important than the other three sub-factors. Top management support makes it easier to implement the goals and strategies of the organization in the direction of GE.

### 5.2. Discussion about ISM Results

To identify the relationships between 20 sub-enablers for GE in manufacturing SMEs, the ISM method has been used in this study. The final ISM model extracted from experts' opinions led to a seven-level hierarchical structure. Here, the hierarchical levels of the model are interpreted from the highest driving power to the lowest. Based on the model, strategic planning and top management support are considered the most basic enablers for GE in SMEs from the point of view of experts. Since these factors have the maximum driving

power, they should be the priority of the attention of senior managers in SMEs. Soewarno et al. have emphasized the high importance of strategic planning for GE. Moreover, Hwang et al. have confirmed strong top-management support as one of the critical success factors for green business parks [69]. The sixth level of the model includes CSR motivation. This factor deserves attention as one of the important factors in GE with high driving power. As a matter of fact, strengthening the motivation of CSR in the organization by focusing on internal and external motivational factors to pursue the goals of GE can increase the participation of employees and managers of different levels of the organization in activities related to GE. The sub-enablers of customer focus, green information systems, and information and analysis are located at the third level, which can develop the fourth level, namely knowledge management practices (knowledge sharing, knowledge creation, and knowledge acquisition). This result revealed that KMPs are directly affected and strengthened through the focus of SMEs on the needs and demands of customers and the use of green information systems and appropriate analytical methods, which has already been partially mentioned in the extant literature [71].

Process management, cross-functional collaboration, green teams, and human resource management are the enablers of the fourth level. Examining the relationships of this level with the lower and higher levels indicated that KMPs can reinforce cooperation for the implementation of GE activities, process management, and human resource management. On the other hand, strengthening the enablers of this level can elevate green purchasing, internal environmental management, investment recovery, and eco-design activities. This way, the development and improvement of green purchasing, internal environmental management, investment recovery and eco-design can facilitate the achievement of social, economic, and environmental dimensions of social responsibility goals. Considering that economic, social, and environmental dimensions are located at the highest level in the model, it can therefore be concluded that the achievement of SCR goals in terms of different dimensions depends on reinforcing enablers that are located in the other six levels.

### 5.3. Discussion about MICMAC Analysis Results

In the last analysis performed on the sub-enablers of GE, analyzing the simultaneous distribution of driving and dependence powers and comparing them with the middle of both scales, the position of sub-enablers was determined in terms of independent, dependent, or linking roles. In effect, the results of this analysis led to the clustering of sub-enablers.

In the first cluster as an independent one, sub-enablers of GE that had weak dependence power and high driving power were placed. The results revealed that nine sub-enablers, including senior management support, strategic planning, analysis and information, green information systems, knowledge sharing, and knowledge creation, are located in this cluster. These enablers actually are crucial for GE, which should be given special attention.

In the second cluster as linkage, sub-enablers had driving and dependence power higher than the scale's middle. Based on the results, four enablers including human resources management, process management, green teams, and cross-functional collaboration were identified as linkage enablers. In fact, if these enablers reinforce, they can develop other enablers, and on the other hand, their successful implementation also depends on independent enablers.

Sub-enablers of the third cluster as dependent ones had higher dependence power and lower driving power than the middle of the scales. The results showed that seven enablers of investment recovery, environmental design, green purchasing, internal environmental management, economic, environmental, and social dimensions (of SCR dimensions) have been positioned in this cluster. This result indicates that the successful implementation of the enablers of this cluster depends on the appropriate development of other sub-enablers.

The factors with weak driving and dependence power were classified as autonomous ones that are not related to the system and do not affect GE in SMEs. The results showed that

none of the sub-enablers was included in this cluster; that is, all the sub-enablers identified in this research are important for GE in SMEs and have an important participation in this phenomenon.

*5.4. Conclusions and Implications*

Today, global concerns related to sustainability and environmental issues, as well as existing competitive pressure, have forced organizations to pay attention to GE and CE as an undeniable necessity [72–74]. Despite the noteworthiness of GE for organizations, especially SMEs, and the awareness of many managers, changing the direction towards GE depends on the availability of special conditions. These special conditions are the enablers that can be analyzed from different organizational, commercial, technological, social, and environmental perspectives [12]. At the same time, organizational enablers are of great importance for intending the approach towards GE [12,73]. Given the internal conditions of the organization are among the controllable factors, therefore, the identification of the enablers can provide practical solutions for managers.

In this study, after identifying organizational enablers in the form of 5 main enablers and 20 sub-enablers, they were prioritized in terms of their importance. The results highlight directing attention to the important enablers by managers and organizational policymakers. According to the experts' opinions in this research, the three main enablers are TQM, CSCM, and CSR. Consequently, strengthening and improving these three enablers should be the top priorities of the organizations. Moreover, among the sub-enablers, including strategic planning, green purchasing, and CSR motivation are prioritized for attention and improvement, respectively.

The model extracted from ISM analysis emphasized the mechanism of improving organizational enablers. Therefore, the managers of SMEs in Iran should prioritize the crucial infrastructure enablers because they are fundamental ones. As a result, the senior managers and policy makers of SMEs should formulate their goals and strategies focusing on GE and CE. In addition, before dealing with knowledge management practices, managers should centralize improving the quality of various information systems and the mechanism of evaluating the information. Furthermore, to strengthen the activities of the circular supply chain, improving cross-functional and team collaboration, as well as process management and human resource management are essential. Senior managers of SMEs should note that achieving results related to social responsibility, which is the core of GE and CE, can only be achieved when special attention is paid to the other enablers that are in the lower levels of the model.

The results of the MICMAC analysis also contain similar suggestions for managers. The results highlighted the importance of the support of senior management, strategic planning, analysis and information, green information systems, knowledge sharing, and knowledge creation, which have high driving power. After that, the improvement of linkage enablers, namely human resources management, process management, green teams, and cross-functional collaboration is the practical suggestion. Finally, all managers' efforts to reinforce independent and linkage enablers can improve investment recovery, environmental design, green purchasing, internal environmental management, and SCR dimensions.

This study identified organizational enablers for GE in the CE, prioritized these factors, identified mutual relationships, and also determined their positions. The study results guide active managers in SMEs in implementing GE in Iran. However, there are some limitations in this research. Considering that organizational enablers were extracted from the literature in a limited period (2010–2023), some enablers out of this period were not included. Therefore, it is suggested that future researchers identify and study other possible enablers. Considering the position of sub-enablers had been determined based on the experts' points of view and with a qualitative approach, it seems necessary to conduct other research using quantitative tools such as exploratory factor analysis to confirm the classification presented in this research. Moreover, to validate the identified relationships

between enablers, it is suggested that future researchers evaluate this mechanism using other techniques such as structural equation modeling. In this study, possible scenarios for the future were not targeted and therefore the issue of uncertainty was not analyzed. It is suggested that future research address this issue.

**Author Contributions:** Conceptualization, M.S., E.M., M.H.B. and A.S.; Methodology, M.S.; Software, A.S.; Validation, M.S.; Formal analysis, E.M.; Investigation, M.H.B.; Data curation, E.M.; Writing—original draft, M.S., M.H.B. and A.S. All authors have read and agreed to the published version of the manuscript.

**Funding:** This research received no external funding.

**Institutional Review Board Statement:** Not applicable.

**Informed Consent Statement:** Not applicable.

**Data Availability Statement:** Not applicable.

**Conflicts of Interest:** The authors declare no conflict of interest.

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
