# Peer review of "Evaluating the Enablers of Green Entrepreneurship in Circular Economy: Organizational Enablers in Focus"

_sustainability, doi:10.3390/su151411253_

Round 1

Reviewer 1 Report

The subject of the article is within the scope of the journal, and it is a new and original contribution. The title of this paper clearly and sufficiently reflects its contents. The abstracts and keywords are informative.

Please, motivate more the abstract, trying to be more concise. Why this work is necessary?

Replace ArXiv papers (if any) (unless very related to the article research area) with related articles from high impact factor journals. These articles from arXiv are not reviewed and therefore you must look for reviewed journal related work from www.sciencedirect.com

In the introduction, you need to connect the state of the art to your paper goals. Please follow the literature review by a clear and concise state of the art analysis. This should clearly show the knowledge gaps identified and link them to your paper goals.

Literature Review has the chance to be further improved: it seems that the authors have made the retrospection. However, via the review, what issues should be addressed? Also please compare your paper with: . A systematic literature review of MABAC method and applications: An outlook for sustainability and circularity. Informatica A decentralized supply chain planning model: a case study of hardboard industry. The International Journal of Advanced Manufacturing Technology,  An integrated machine learning and quantitative optimization method for designing sustainable bioethanol supply chain networks. Decision Analytics Journal, A fuzzy sustainable model for COVID-19 medical waste supply chain network. Fuzzy Optimization and Decision Making, A sustainable-circular citrus closed-loop supply chain configuration: Pareto-based algorithms. Journal of Environmental Management, A New Wooden Supply Chain Model for Inventory Management Considering Environmental Pollution: A Genetic algorithm. Foundations of Computing and Decision Sciences, Designing a sustainable bioethanol supply chain network: A combination of machine learning and meta-heuristic algorithms. Industrial Crops and Products,  Designing a sustainable closed-loop supply chain network considering lateral resupply and backup suppliers using fuzzy inference system. Environment, Development and Sustainability,  Designing a new multi-echelon multi-period closed-loop supply chain network by forecasting demand using time series model: a genetic algorithm. Environmental Science and Pollution Research

The assumptions of the model are not clearly justified.

Research limitations can help researchers overcome obstacles. Please add research limitations to the conclusion section.

The writing of the paper needs an improvement in terms of grammar, spelling, and presentations.

The writing of the paper needs an improvement in terms of grammar, spelling, and presentations.

Author Response

Point 1: Please, motivate more the abstract, trying to be more concise. Why this work is necessary?

Response 1: The abstract is modified.

Point 2: In the introduction, you need to connect the state of the art to your paper goals. Please follow the literature review by a clear and concise state of the art analysis. This should clearly show the knowledge gaps identified and link them to your paper goals.

Response 2: The introduction and literature modified.

Point 3: Literature Review has the chance to be further improved: it seems that the authors have made the retrospection. However, via the review, what issues should be addressed? Also please compare your paper with: A systematic literature review of MABAC method and applications: An outlook for sustainability and circularity. Informatica, A decentralized supply chain planning model: a case study of hardboard industry. The International Journal of Advanced Manufacturing Technology, an integrated machine learning and quantitative optimization method for designing sustainable bioethanol supply chain networks. Decision Analytics Journal, A fuzzy sustainable model for COVID-19 medical waste supply chain network. Fuzzy Optimization and Decision Making, A sustainable-circular citrus closed-loop supply chain configuration: Pareto-based algorithms. Journal of Environmental Management, A New Wooden Supply Chain Model for Inventory Management Considering Environmental Pollution: A Genetic algorithm. Foundations of Computing and Decision Sciences, Designing a sustainable bioethanol supply chain network: A combination of machine learning and meta-heuristic algorithms. Industrial Crops and Products, Designing a sustainable closed-loop supply chain network considering lateral resupply and backup suppliers using fuzzy inference system. Environment, Development and Sustainability, Designing a new multi-echelon multi-period closed-loop supply chain network by forecasting demand using time series model: a genetic algorithm. Environmental Science and Pollution Research.

Response 3: The literature review by using the recommended articles improved.

Point 4: The assumptions of the model are not clearly justified.

Response 4: The most important assumption of ISM is the transitivity principle which is explained in the third part of the methodology.

Point 5: Research limitations can help researchers overcome obstacles. Please add research limitations to the conclusion section.

Response 5: Research limitations were added to the conclusion section.

Point 6: The writing of the paper needs an improvement in terms of grammar, spelling, and presentations.

Response 6: English/grammatical errors throughout the text were corrected.

Reviewer 2 Report

This work tries to assess the enablers of green entrepreneurship in CE by proposing an MCDM-based approach. The idea is interesting but the manuscript needs to be improved.

a. Abbreviations should be spelled out both in the abstract and main text.

b. Introduction section: here, you should define the problem, address the significance, review the applications, and delineate the contributions and research goals.

c. How can you address uncertainty within your decision making framework?

d. The validation of the results as well as comparative analysis should be provided.

e. Any sensitivity analysis section to help managers learn better about the role of parameters in decision-making?

f. Why BWM as the MCDM? for example why not ANP?

There are some English/grammatical errors in the text. The authors must review the English/writing of the text.

Author Response

Point 1: Abbreviations should be spelled out both in the abstract and main text.

Response 1: The mentioned corrections were made.

Point 2: Introduction section: here, you should define the problem, address the significance, review the applications, and delineate the contributions and research goals.

Response 2: The introduction is improved.

Point 3: How can you address uncertainty within your decision-making framework?

Point 4: The validation of the results as well as comparative analysis should be provided.

Point 5: Any sensitivity analysis section to help managers learn better about the role of parameters in

decision-making?

Responses 3, 4, 5: The validation in section 4.3 was added to the article and explanations related to 3, 4 and 5 points were provided.

Point 6: Why BWM as the MCDM? for example why not ANP?

Response 6: The most crucial advantage of BWM as one of the MCDM techniques compared to the AHP and ANP is more accuracy in the comparisons due to the reduction of the total number of comparisons and the decrease in inconsistencies in the results.

Point 7: here are some English/grammatical errors in the text. The authors must review the English/writing of the text.

Response 7: English/grammatical errors throughout the text were corrected.

Reviewer 3 Report

Review report. Minor corrections.

The manuscript “Evaluation the Enablers of Green Entrepreneurship in Circular Economy: Organizational Enablers in Focus” analyzes organizational enablers for green entrepreneurship in manufacturing SMEs in Iran. Organizational enablers were extracted by reviewing the literature and then, using the viewpoints of active experts in different industries in SMEs, they were classified. In the next step, the "Best Worst Method" was employed to prioritize the identified enablers (5 factors) and sub-enablers (20 factors). Using the "Interpretive Structural Modeling" method, the contextual hierarchical relationships between these factors were identified and the corresponding model was presented. Using MICMAC analysis, driving and dependence powers of organizational enablers have been calculated and the enablers were clustered

This review study deals with interesting and very up-to-date theme. The results/conclusions are useful for solving actual ‘real life’ problems regarding CE entrepreneurship. The study is scientifically sound and relevant for the field. The text is clear. Presentation is well-structured.

The paper is appropriate for this Special Issue of Sustainability Journal.  

Abstract is informative as it covers all of the main points of the work.

‘Green Entrepreneurship’ and ‘Circular Economy’ can be removed from the keywords as these terms are already mentioned in the title.

Introduction and Literature Review are consistent and informative. Introduction is brief. The provided review of literature is clear, comprehensive and of relevance to the field. Authors identified the gap in knowledge and highlighted it. There is no need for abridging nor changing the text.

Materials and methods chapter is clear and understandable. However, I would suggest changing of the chapter title into ‘Methodology’ since there is no traditional experimental work in this study.

Results chapter and the following Discussion chapter provide a clear presentation of correlated results. The discussion is scientifically sound. The experimental design (i.e. analytical modeling employed) is appropriate to test the hypothesis as there are no identified errors of fact and logic.

The conclusions are adequate.  However, I would suggest placing them in separate chapter to increase their visibility.   

The cited references are appropriate and up-to-date (within the last 5 years or so). Used literature is relevant. There is no excessive number of self-citations.

English language is good; however, the text should be read one more time to exclude all typing mistakes.

Author Response

Point 1: ‘Green Entrepreneurship’ and ‘Circular Economy’ can be removed from the keywords as these terms are already mentioned in the title.

Response 1: ‘Green Entrepreneurship’ and ‘Circular Economy’ are removed from the keywords.

Point 2: Introduction is brief. 

Response 2: The Introduction is improved.

Point 3: Materials and methods chapter is clear and understandable. However, I would suggest changing of the chapter title into ‘Methodology’ since there is no traditional experimental work in this study.

Response 3: the chapter title changed to ‘Methodology’.

Point 4: The conclusions are adequate. However, I would suggest placing them in separate chapter to increase their visibility.

Response 4: Conclusions and implications are presented in section 5.4 according to the journal's format.

Reviewer 4 Report

Dear Authors

The comments requiring your kind attention are found in the pdf attachment. Please kindly use Microsoft Edge to open the pdf.

Please kindly thoroughly go through English grammar and sentence structures.

Thank you.

Dear Authors

It appears to me there is frequent usage of the present perfect and past perfect tenses. In my humble opinion, in many instances, you could use the past tense.

Some journals recommend the present tense unless the past tense is clearly required.

I leave tenses to your wisdom.

Thank you.

Author Response

Point 1: The comments requiring your kind attention are found in the pdf attachment. Please kindly use Microsoft Edge to open the pdf.

Response 1: All corrections mentioned in the file were corrected. The corrections include the followings:

  1. The abstract was modified.
  2. The introduction and the literature improved.
  3. The Delphi method was completed.
  4. English/grammatical errors throughout the text were corrected.

Round 2

Reviewer 2 Report

Good work.

Good work.

Author Response

Thank you for your kind feedback. We asked a native English language proofreader to check the revised draft.